# Canine tumor mutational burden is correlated with *TP53* mutation across tumor types and breeds

Burair A. Alsaihati [1,2], Kun-Lin Ho[1], Joshua Watson[1], Yuan Feng[1], Tianfang Wang[1], Kevin K. Dobbin[3] & Shaying Zhao [1]✉

Spontaneous canine cancers are valuable but relatively understudied and underutilized models. To enhance their usage, we reanalyze whole exome and genome sequencing data published for 684 cases of >7 common tumor types and >35 breeds, with rigorous quality control and breed validation. Our results indicate that canine tumor alteration landscape is tumor type-dependent, but likely breed-independent. Each tumor type harbors major pathway alterations also found in its human counterpart (e.g., PI3K in mammary tumor and p53 in osteosarcoma). Mammary tumor and glioma have lower tumor mutational burden (TMB) (median < 0.5 mutations per Mb), whereas oral melanoma, osteosarcoma and hemangiosarcoma have higher TMB (median ≥ 1 mutations per Mb). Across tumor types and breeds, TMB is associated with mutation of *TP53* but not *PIK3CA*, the most mutated genes. Golden Retrievers harbor a TMB-associated and osteosarcoma-enriched mutation signature. Here, we provide a snapshot of canine mutations across major tumor types and breeds.

[1] Department of Biochemistry and Molecular Biology, Institute of Bioinformatics, University of Georgia, Athens, GA, USA. [2] National Center for Genomics Technology, King Abdulaziz City for Science and Technology, Riyadh, Saudi Arabia. [3] Department of Epidemiology and Biostatistics, University of Georgia, Athens, GA, USA. ✉email: szhao@uga.edu

Cancers in pet dogs arise spontaneously in animals that have intact immune systems and share the same environment as humans. Compared to traditional cancer models such as cell lines and rodents, these canine cancers more accurately emulate human cancers in etiology, complexity, heterogeneity, behavior, treatment and outcome. Hence, they have the potential to effectively bridge a current gap between preclinical studies and human clinical trials, accelerating bench-to-bedside translation[1–3]. As such, the National Cancer Institute (NCI) has recently issued programs targeting canine cancers. These include funding multi-institute immunotherapy trials in pet dogs and a 5-year project to build the NCI Integrated Canine Data Commons, a database for canine data dissemination similar to the cancer genome atlas (TCGA) data portal. Private foundations are also funding canine studies, including the Vaccination Against Canine Cancer Study, a 5-year, $6 million trial to vaccinate 800 healthy dogs using tumor-specific neoantigens to determine if the vaccination will prevent or delay the onset of cancer.

However, current deficiencies create roadblocks to the effective use of canine cancers. This is clearly exemplified by sequence mutation, a hallmark of cancer[4]. Mutation landscape, burden and signature have all been extensively investigated in human cancer via pan-cancer studies[5–10]. However, to our knowledge, no pan-cancer research has been published for the dog and fundamental questions remain unanswered. For example, does the canine tumor mutation landscape match that of human cancer? Does canine tumor mutational burden (TMB) also vary significantly among cancer types, as it does in human cancers[5,6]?

The lack of pan-breed cancer study also leaves key questions unanswered. For example, Golden Retrievers are predisposed to the development of osteosarcoma, lymphoma and hemangiosarcoma; do the mutation landscape and TMB of Golden Retriever differ among these cancer types? Golden Retriever, Greyhound, and Rottweiler dogs are all predisposed to osteosarcoma; do the mutation landscape and TMB of osteosarcoma differ among these breeds? Addressing these questions will significantly enhance the usage of >300 pure breeds of the dog in cancer research.

Here we show that a pan-tumor and pan-breed analysis may answer some of these questions. Our study consists of matched tumor and normal samples of 684 cases, which represent over 7 common canine tumor types and over 35 popular breeds, with published whole-exome sequencing (WES)[11–19] (654 cases) and/ or whole-genome sequencing (WGS) data[13,17,20] (86 cases). A total of 600 cases have passed our comprehensive sequence quality controls (QC), of which 440 cases have their breeds validated, corrected, or predicted using breed-specific germline base substitutions and small indels discovered for 10 pure breeds. We have then investigated somatic mutations, which include somatic base substitutions and small indels, as well as gene amplification and deletions in these canine cases. The results indicate that these alterations are tumor type-dependent, but mostly breed-independent. Across tumor types and breeds, TMB, defined as the number of somatic base substitutions and small indels per Mb callable coding sequence (CDS), is associated with mutation of TP53 but not PIK3CA, the two most mutated genes. Finally, each tumor type harbors major pathway alterations that are also found in its human counterpart. Our study provides a snapshot of mutations across major tumor types and breeds in pet dogs.

## Results

### QC of published canine sequencing data.
The WES dataset consists of 1316 paired tumors and normal samples of 654 animals from 8 BioProjects (Supplementary Data 1). These include 204

cases (408 samples) of mammary tumor[11,12], 56 cases (112 samples) of glioma[13], 61 cases (122 samples) of B-cell lymphoma[14], 39 cases (78 samples) of T-cell lymphoma[14], 65 cases (136 samples) of oral melanoma[15], 78 cases (156 samples) of osteosarcoma[16,17], 68 cases (138 samples) of hemangiosarcoma[18,19] and 83 cases (166 samples) of unclassified tumors (Supplementary Data 1). They represent over 35 breeds, including Golden Retriever (163 dogs), Maltese (69 dogs), Poodle (38 dogs), Boxer (36 dogs) and others listed in Supplementary Data 1.

One of the mammary tumor studies[11] provides the most comprehensive case information (Supplementary Data 1), with the patient (e.g., age, sex and breed), histological subtype and limited clinical (e.g., tumor invasiveness and patient alive/death status) data. The osteosarcoma, lymphoma, glioma and hemangiosarcoma studies all have patient information (Supplementary Data 1) but lack clinical data. The oral melanoma study lacks patient information, including breed.

The WES data were generated by different groups, using different exome-capturing kits and Illumina sequencing machines. We hence performed a rigorous QC to ensure that data chosen from each study meet a set of quality standards before any integrative analysis.

For the sequencing amount, except for certain mammary and hemangiosarcoma sample sets, all datasets have a median of >50 million (M) read pairs per sample (Fig. 1a). We excluded two samples with <5 M read pairs from further analyses (Supplementary Data 1).

We then examined the mapping of read pairs to the canine reference genome[21]. Except for the glioma and one hemangiosarcoma datasets, all studies have >80% read pairs in nearly every sample uniquely and concordantly mapped to the genome, with the median close to or larger than 90% (Fig. 1b). We excluded 9 samples with mapping rates <60% (Supplementary Data 1). Furthermore, except for glioma and hemangiosarcoma, nearly all samples have >70% reads (close to 90% for mammary and melanoma samples) with a mapping quality score of >30 (Fig. 1c). For the target rate, all studies except two have, on average, >50% read pairs that are uniquely and concordantly mapped to the CDS regions, with the melanoma study and one mammary tumor study[11] achieving >60% (Fig. 1d). We excluded three samples with target rates <30% (Supplementary Data 1). For the average mapped read coverage in CDS regions, except for a hemangiosarcoma dataset[19], all studies have reached a median of >70X (Fig. 1e). We excluded 24 samples with coverage <30X (Supplementary Data 1). For the mapped read distribution in the target regions (which reflects sequencing randomness), we determined the deviation of each sample from its theoretical Poisson distribution (as a completely random sequencing process can be approximated by the Poisson distribution). The results indicate that one mammary tumor study[11] has the most random sequencing, closely followed by the oral melanoma study (Fig. 1f). We excluded one sample which is a clear outlier (Fig. 1f; Supplementary Data 1). After these steps, all samples have >10 Mb callable bases in total in CDS regions (used for somatic base substitution and small indel discovery; see "Methods") (Fig. 1g).

To assess the tumor-normal sample-pairing accuracy, we used germline base substitution and small indel variants detected in each sample, assuming that correctly paired samples, compared to other samples in the same study, should share the most variants. We found a total of 24 mispaired cases (Fig. 1h) and excluded them from further analysis (Supplementary Data 1).

In summary, our QC analysis indicates that one of the mammary studies[11] and the oral melanoma study[15] have the highest sequence quality and that the mammary study[11] has the most comprehensive case information. A total of 591 cases (597 tumors and 591 matching normal samples) have passed our

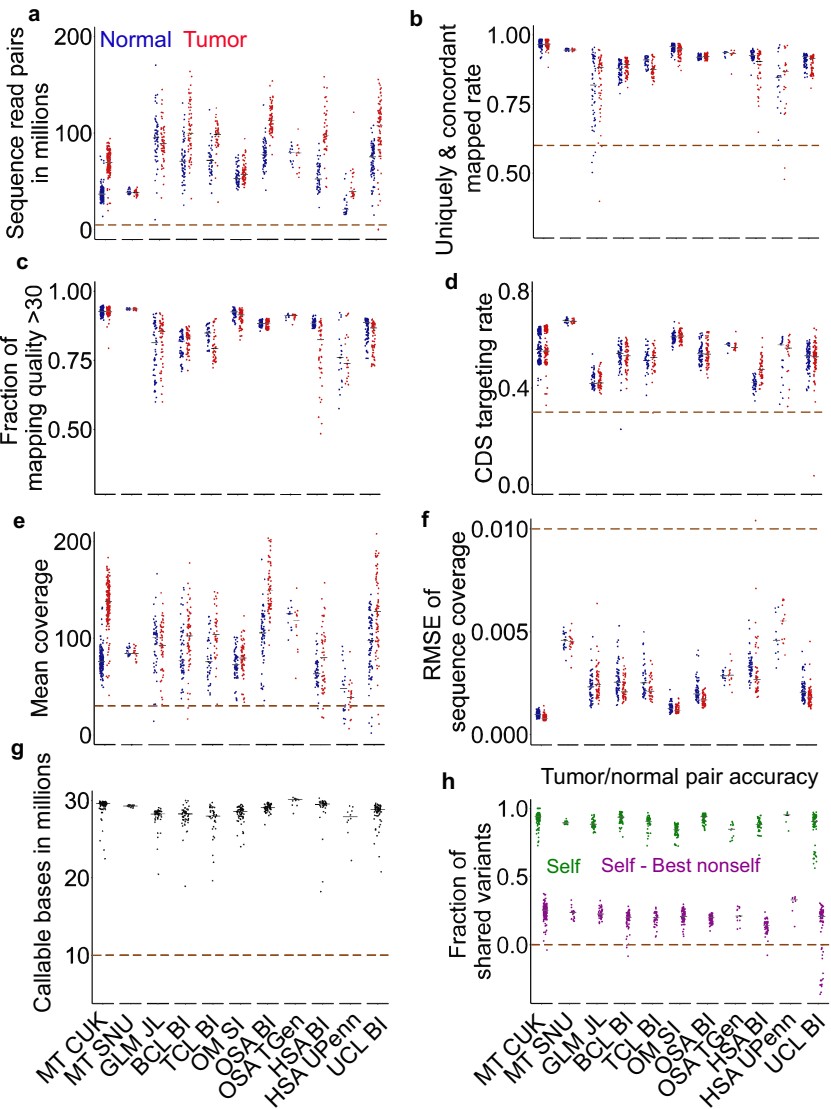

**Fig. 1 We performed a rigorous quality control (QC) of whole-exome sequencing (WES) data published for 654 canine cases. a** Distributions of total read pairs per sample of the tumor and normal sample sets of each study. Each dot represents a sample and the median is indicated by a black line. The dashed line specifies the QC cutoff. Each study is represented by the tumor type and the institute name. MT mammary tumor, GLM glioma, BCL B-cell lymphoma, TCL T-cell lymphoma, OM oral melanoma, OSA osteosarcoma, HSA hemangiosarcoma, UCL unclassified. CUK Catholic University of Korea, SNU Seoul National University, JL Jackson Laboratory, SI Sanger Institute, BI Broad Institute, TGen Translational Genomics Research Institute, UPenn University of Pennsylvania. n = 184, 20, 56, 61, 39, 65 (71 tumors), 66, 12, 47, 21 (23 tumors), and 83 independent cases for matched normal and tumors samples for each independent study listed from left to right. **b–f** Distributions of per sample rate of read pairs that aligned concordantly and uniquely to the canFam3 reference genome (**b**) (n = 81 for UCL BI; others the same as **a**), the fraction of reads with mapping quality of ≥30 (**c**) (n = 50 and 18 for GLM JL and HSA UPenn respectively; others the same as **b**), CDS-targeting rate (the fraction of read pairs that align concordantly and uniquely to the canFam3 CDS regions) (**d**) (the same sample size as **c**), mean read coverage in CDS regions (**e**) (n = 60, 38 and 80 for BCL BI, TCL BI, and UCL BI, respectively; others the same as **d**) and root-mean-square error (RMSE) between the actual distribution and theoretical distribution (based on the Poisson distribution) of sequence coverage in CDS regions (**f**) (n = 183, 49, 58, 43, 8, and 74 for MT CUK, GLM JL, BCL BI, HSA BI, HSA UPenn, and UCL BI, respectively; others the same as **e**). **g** Distributions of the total number of callable bases per case, determined by MuTect. n = 183, 20, 49, 58, 38, 71, 66, 12, 42, 8, and 74 independent tumors from left to right. **h** Tumor-normal pairing accuracy. "Self" (in green) is the fraction of germline variants shared between the normal and tumor samples of a dog. "Best nonself" is the fraction of germline variants shared between a normal or tumor sample of one dog and its best-matched sample from another dog. "Self—Best nonself" (in purple) indicates the difference and a negative difference points to incorrect tumor-normal pairing. The sample size is the same as in (**g**). Source data are provided as a Source Data file.

QC measures (Supplementary Data 1) and were used for further analyses.

We also performed similar QC analyses on the WGS dataset, which consists of 172 paired tumor and normal samples from 86 animals with glioma (67 cases)[13], oral or ocular melanoma (4 cases)[20], or osteosarcoma (15 cases)[17] (Supplementary Data 1).

Close to 30 breeds are covered, including Boxer (24 animals), Boston Terrier (11 animals), and others listed in Supplementary Data 1. We found 25 samples with a mapping rate <60% and 25 samples with a sequence coverage <30X (Supplementary Fig. 1; Supplementary Data 1) and excluded them from further analysis. Because of the small sample size (only 72 paired tumors and

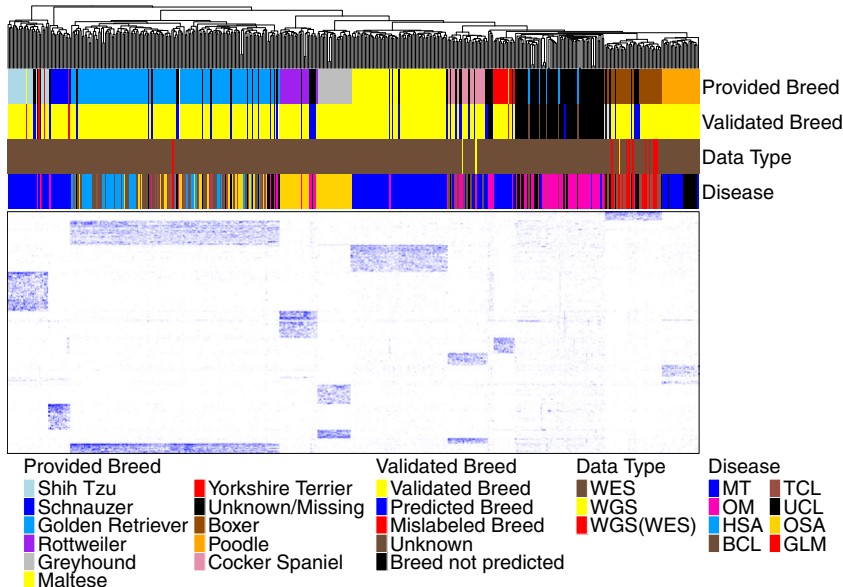

**Fig. 2 We conducted breed validation and prediction using breed-specific germline base substitutions and small indels discovered.** The heatmap shows the clustering of 505 animals (398 dogs with breeds provided and 107 dogs with no breeds provided), using variant allele frequency (VAF) values of the 5363 breed-specific germline base substitution and small indel variants in their normal samples. These variants were discovered with the WES dataset (see "Methods"). The WGS dataset was used for validation as specified in the "Data Type" bar, where "WGS(WES)" indicates that a dog has both WGS and WES data but only WGS data were used in the clustering analysis. The "Provided Breed" bar and the "Disease" bar respectively indicate the breed and tumor type of each dog provided by the source studies. The "Validated Breed" bar denotes the analysis outcome as specified, with "Unknown" representing dogs whose provided breeds could not be validated or corrected, due to the lack of any specific VAF clustering patterns of the ten pure breeds investigated. Source data are provided as a Source Data file.

normal samples from 36 cases passed QC) (Supplementary Data 1), we used the WGS dataset only for breed validation and noncoding mutation signature finding.

**Breed-specific germline analysis for breed validation.** To assess the breed data accuracy, we focused on the 10 pure breeds in the WES dataset with each having ≥10 animals passing QC measures specified in Fig. 1, and developed a breed validation and prediction software (Supplementary Software 1). Briefly, we identified 5363 breed-specific variants (Supplementary Data 2), defined as germline base substitutions and small indels that are unique to or enriched in one of these breeds. We then performed clustering analysis using the variant allele frequency (VAF) values of these variants in the normal samples of the animals. Our analysis validated the breeds of 385 dogs and corrected 5 dogs with breed error (3 Yorkshire Terriers reassigned to 2 Shih Tzus and 1 Schnauzer; 1 Maltese each reassigned to Shih Tzu and Yorkshire Terrier) (Fig. 2). We also reclassified 5 dogs as "unknown", as they lack VAF patterns seen in any of the 10 pure breeds (Fig. 2).

To corroborate our strategy, we first performed the same clustering analyses using the WGS dataset after QC specified in Supplementary Fig. 1. As shown in Fig. 2 and Supplementary Fig. 2, all 22 dogs (3 having WGS data only), whose reported breeds belong to one of the 10 pure breeds investigated above, were confirmed. Second, we divided the WES studies into discovery and validation sets based on their sample size (Supplementary Fig. 2a). We identified breed-specific germline variants for 9 pure breeds with ≥10 animals per breed in the discovery set (Supplementary Data 2), with which we clustered dogs from both sets. The analysis confirmed 17 of 19 animals from the validation set and reassigned the breed for the remaining 2 dogs (Supplementary Fig. 2a). These results indicate that our approach is valid.

We repeated this analysis to attempt breed prediction for 107 cases in the WES dataset with no breed data (e.g., oral melanoma

cases[15]). We were able to unambiguously assign breeds to 50 dogs (14 to Golden Retriever, 10 to Cocker Spaniel, 8 each to Boxer and Rottweiler, 4 each to Shih Tzu and Maltese, and 1 each to Yorkshire Terrier and Schnauzer) (Fig. 2).

Lastly, we clustered all 626 animals with WES and/or WGS data passing QC (Fig. 1 and Supplementary Fig. 1), including 85 dogs with reported breeds not among the 10 pure breeds investigated (other breeds), as well as 24 dogs of mixed breed. We hypothesize that if our approach is valid, the vast majority of these dogs would not cluster with the 10 pure breed dogs. Our analysis classified 18 mixed breed dogs as "unknown" and reassigned the remaining 6 dogs to specific pure breeds (2 Maltese, 2 Schnauzer, 1 Rottweiler, and 1 Shih Tzu) (Supplementary Fig. 2c). For 85 dogs of other breeds, the analysis classified 82 dogs as "unknown" and reassigned the breed for the remaining 3 dogs (Supplementary Fig. 2c). All other dogs shared the same breed validation, correction, prediction, and reclassification indicated in Fig. 2. The results support our hypothesis, indicating that our approach is effective.

In summary, we discovered breed-specific germline variants for 10 pure breeds, with which we successfully validated 385 dogs, corrected 5 dogs and predicted 50 dogs in the WES dataset for their breed assignment, as shown in Fig. 2. These dogs were used for downstream breed-related analyses described later.

**Alteration landscape varies with tumor types but not breeds examined.** For somatic mutations (i.e., base substitutions and small indels), we focused on the WES dataset, because of the large sample size (597 tumor-normal pairs from 591 cases after QC), and high sequencing coverage (Fig. 1; Supplementary Data 1) and the CDS regions, which are more accurately annotated than other genomic regions. We assembled a mutation discovery pipeline that used sequence coverage, mutant allele frequency (MAF) and paired-read strand orientation[22] to reduce mutation artifacts (5-step filtering; see "Methods"). This effectively reduces C > T

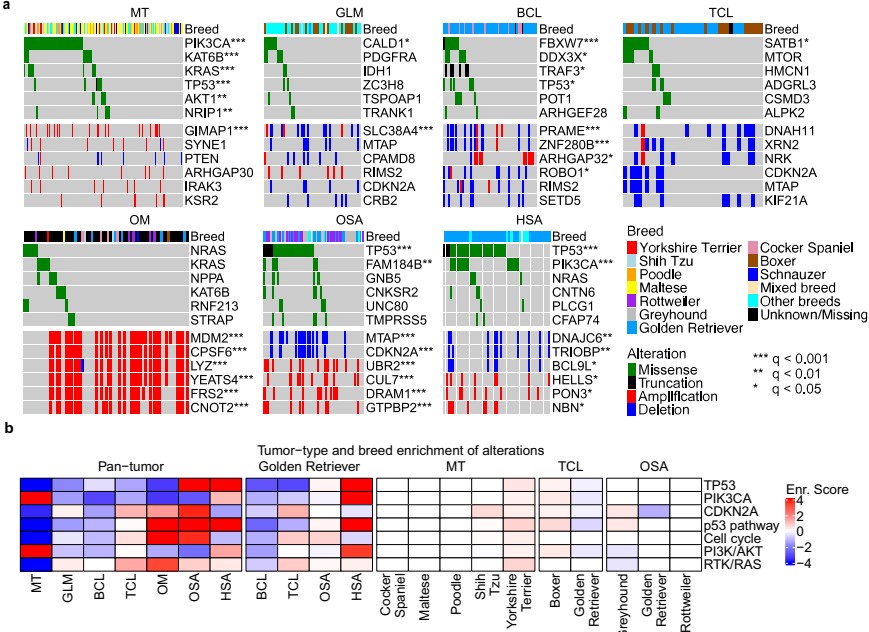

**Fig. 3 Canine tumor alteration landscape, consisting of genes recurrently mutated and/or amplified/deleted, varies with tumor types but not with breeds in general. a** Oncoprints indicate the top six most recurrently altered genes with nonsynonymous somatic base substitutions or small indels (top), or copy number alterations (CNAs) (bottom), in CDS regions in each tumor type indicated. Significant recurrence, identified by Fisher exact tests with multiple testing correction[60], are denoted by "*" as shown. The breed of each animal is specified in the breed bar. **b** Enrichment scores of the most recurrently altered genes and pathways, obtained via Fisher exact test q-values (see "Method"), in each tumor type of all breeds (left) and of Golden Retriever (middle), as well as in each breed with ≥10 dogs within a tumor type (right). Source data are provided as a Source Data file.

artifacts originated from the fixation process in FFPE samples[23], as well as G > T artifacts arisen from 8-oxoG DNA oxidative damage[22] in frozen samples of certain studies (Supplementary Fig. 3).

We compared each mutation in each tumor between our study and the original publications, including the genomic coordinate and the actual mutation, which are published only for the mammary tumor[11] and oral melanoma[15] studies. For oral melanoma, we found a median overlap rate of 67% with 5-step filtering and of 59% with further paired-read strand orientation filtering (Supplementary Fig. 4). We manually examined >20 mutations detected only by our pipeline or in the original publications and found that all appear to be valid base changes (a few examples provided in Supplementary Fig. 4). Thus, the difference is likely due to variations in read cleaning, germline mutation filtering, and artifact filtering. For mammary tumors, the overlap rate is lower (43%) (Supplementary Fig. 4) due to different mutation calling software, as 66% overlap was achieved when we used MuTect2 as in the original publication[11] (Supplementary Fig. 4).

We identified genes that harbor somatic non-synonymous base substitutions or small indels, as well as genes that are amplified or deleted, in each tumor (Supplementary Data 3). We then examined the alteration landscape (Fig. 3a), which consists of these altered genes that can be detected at ≥0.8 power within a tumor type or a breed based on our sample size calculation (Supplementary Fig. 5a). The study reveals unique alteration features for each canine tumor type (Fig. 3a), many of which are consistent with individual tumor type findings[11–20,24].

Mammary tumors harbor frequent PI3K pathway alteration, with 50% of the tumors having at least one member gene-altered (Fig. 3a; Supplementary Data 3). The PIK3CA H1047R mutation is especially common, found in 26% of the tumors (Supplementary Data 3). However, another PIK3CA mutation hotspot, the

E542/545 site, is intriguingly missing, differing from human breast cancer[25].

Oral melanoma and osteosarcoma both harbor frequent p53 pathway alteration (61%) (Fig. 3a; Supplementary Data 3). However, the actually altered genes differ, with TP53 mutated in 50% of osteosarcomas and MDM2 amplified in 45% of oral melanomas (Fig. 3a, Supplementary Data 3). Moreover, while deletion is common in osteosarcoma, amplification is frequent in oral melanoma (Supplementary Data 3). Indeed, CDKN2A is deleted in 22% of osteosarcomas and CDK4 is amplified in 28% of oral melanomas, resulting in frequent cell cycle gene alteration in both tumor types (Supplementary Data 3).

Hemangiosarcoma has a TP53 mutation frequency of 59%, the highest among the 7 tumor types (Fig. 3a). PIK3CA is another frequently mutated gene, mutated in 31% of hemangiosarcomas. The most significantly mutated genes include FBXW7 (encoding WNT signaling molecule) in B-cell lymphoma, SATB1 (functioning in chromatin remodeling) in T-cell lymphoma, and CALD1 (encoding an actin and myosin binding protein) in glioma (Fig. 3a). However, they are less recurrent than PIK3CA mutation in mammary tumors or TP53 mutation in hemangiosarcoma or osteosarcoma (Fig. 3a).

In contrast to tumor type, canine alteration landscape appears largely breed-independent among the breeds examined (Fig. 3a). To statistically test this, we performed Fisher exact tests on the most recurrently altered genes (TP53, PIK3CA, and CDKN2A) and pathways (p53, PI3K, cell cycle, and RTK/RAS) to achieve a larger power (Supplementary Fig. 3b). Most of these alterations do not differ significantly in their enrichment or depletion levels among different breeds within the same tumor type, unlike the tumor type comparison (Fig. 3b). For example, mammary tumors of Maltese, Shih Tzu, and Yorkshire Terrier dog all have frequent PIK3CA mutation and PI3K pathway alteration (Fig. 3b and Supplementary Fig. 3b). However, various tumor types of Golden

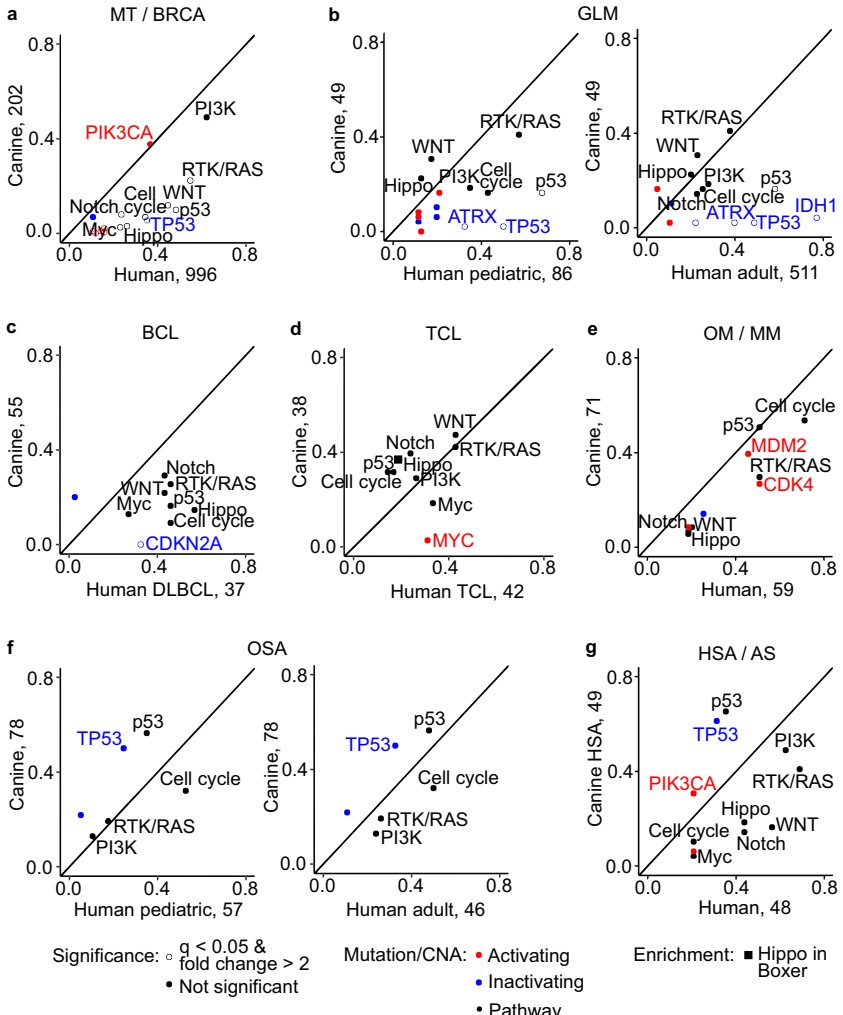

**Fig. 4 Canine tumors share many of the same major gene and pathway alterations as their human counterparts.** Each panel of **a**–**g** compares the mutation recurrence of a gene or pathway in a tumor type between the two species. Human breast cancer (BRCA) (**a**), pediatric and adult GLM (**b**), diffuse large B-cell lymphoma (DLBCL) (**c**), TCL (**d**), mucosal melanoma (MM) (**e**), OSA (**f**) and angiosarcoma (AS) (**g**) are from published studies (see text). Shown are curated pathways (10 total) from a TCGA pan-cancer publication[46], as well as genes mutated in ≥10 tumors and ≥10% (for pathway genes) or 20% (for non-pathway genes) of all tumors in a tumor type in either species. Genes and pathways with the mutation frequency that are significantly different ($q < 0.05$ from Fisher exact tests followed by multiple testing correction[60]) and have ≥2-fold changes between the two species are considered different. Source data are provided as a Source Data file.

Retriever dogs differ significantly in these alterations (Fig. 3b and Supplementary Fig. 3b).

**Canine and human tumors share many major alterations.** We observed numerous dog–human homologies, including significant mutation of the same residues (e.g., PIK3CA H1047 is mutated in >25% of mammary tumors of both species) and genes (e.g., in both species, *TP53* is mutated in >25% osteosarcomas and *MDM2* is amplified in about 40% oral or mucosal melanomas) (Fig. 4). The highest homology, however, is shown at the pathway level (Supplementary Data 4). Significant pathway alterations found in each canine tumor type have also been reported in its human counterpart (Fig. 4), including breast cancer[26,27], pediatric and adult glioma[7,8,27–29], diffuse large B-cell lymphoma[27], T-cell lymphoma[30], mucosal melanoma[31], osteosarcoma[7,8,32] and angiosarcoma[33].

In both species, p53 pathway alteration and cell cycle alteration are equally common in osteosarcoma, oral melanoma and T-cell lymphoma (Fig. 4). However, alteration of individual pathway members may differ. For example, *CDKN2A* is deleted in 22% of canines but only altered in 5% of human pediatric osteosarcomas,

while *RB1* is altered in 16% of human pediatric osteosarcomas but none of the canine tumors (Fig. 4). The opposite trend is observed in T-cell lymphomas for *RB1* (Fig. 4). *CDKN2A* and *RB1* both negatively regulate the cell cycle and deletion of either promotes cell proliferation.

PI3K signaling is the most frequently altered pathway in mammary tumors of both species, with an alteration rate of ≥50% (Fig. 4). *PIK3CA* mutation and *PTEN* deletion are common in both species (Fig. 4). PI3K alteration is also common in gliomas and hemangiosarcomas/angiosarcomas of both species (Fig. 4). The same is true for RTK/RAS signaling.

We also observed several significant differences between the two species. For example, *TP53* mutation and p53 pathway alteration are significantly more common in human breast cancer and glioma than in their corresponding canine tumors (Fig. 4). Chromatin remodelers *IDH1* and *ATRX* are altered in significantly more human gliomas[7,8,27] than in canine gliomas (Fig. 4).

**Canine TMB varies mostly among tumor types but not breeds.** We investigated TMB, defined as the number of somatic base

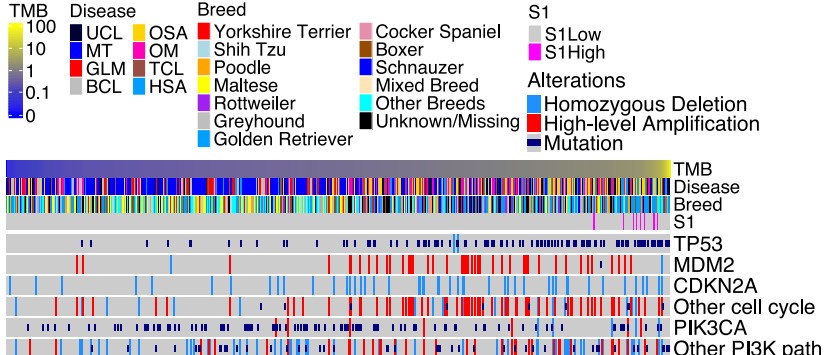

**Fig. 5 We investigated TMB and common alterations in each of the 597 tumors of over 7 tumor types and over 35 breeds.** The tumors in the oncoprint were ordered from left to right by lowest to highest TMB. Seven tumor types as indicated in Fig. 3 and unknown tumor types (UCL; see Fig. 1) are included. Breeds shown include those validated, corrected, predicted or unknown (with an issue or not predicted) as shown in Fig. 2, as well as other breeds, which are not validated due to small sample size, and mixed breeds. Top recurrent gene and pathway alterations are shown. Source data are provided as a Source Data file.

substitutions and small indels per Mb callable CDS, in each of the 597 canine tumors of the WES dataset after sequence QC (Fig. 5). To increase the accuracy, we first identified 1564 retrogenes and other problematic genes (see "Methods") in the current canine gene annotation database (Supplementary Data 5). We excluded these problematic genes from TMB calculations, as they harbor significantly more mutations compared to protein-coding genes (Fig. 6a).

Resembling human cancer[6], TMB varies among these canine tumors, ranging from 0 to 36 (Fig. 5). However, the overall TMB is low, with a median of 0.53 (Fig. 5). Hypermutation (TMB > 10) was found in 1.17% of canine tumors, and ultra-hypermutation (TMB > 100) was not detected in any tumors. Both are rarer than in adult human tumors, where 2.3% are hypermutated and 0.32% are ultra-hypermutated (Fig. 6).

TMB varies among tumor types (Fig. 6a, and Supplementary Figs. 5c and 6). Canine mammary tumors, glioma and B-cell lymphoma have lower TMB, with a median range of 0.37–0.4 and are therefore classified as TMB-low (TMB-L) (Fig. 6a). Canine T-cell lymphoma, oral melanoma, osteosarcoma and hemangiosarcoma have significantly higher TMB, with a median range of 0.81–1.08 and are thus classified as TMB-high (TMB-H) (Fig. 6a). Except for lymphomas (see "Discussion"), these findings are confirmed with different mutation discovery strategies (Supplementary Fig. 7a).

As sequence coverage influences the sensitivity of somatic mutation discovery[34], we performed TMB comparison across tumor types controlling for sequence coverage (at 30–50x, 50–100x, and >100x) (Supplementary Fig. 6a). The analysis confirms our original conclusion that TMB is tumor type-dependent (Supplementary Fig. 6b).

Within the same tumor type, TMB appears to be similar among breeds, except for osteosarcoma where Golden Retrievers have significantly higher TMB than Rottweilers and Greyhounds (Fig. 7a, and Supplementary Figs. 5c and 6c). We thus conclude that canine TMB primarily varies with tumor types, but not breeds for those examined (Fig. 7a, and Supplementary Figs. 5c and 6c).

In general, canine TMB values are significantly lower than their human adult counterparts[9] and are more comparable to their pediatric counterparts[29,32] (Supplementary Fig. 7b).

**Canine TMB is correlated with *TP53* but not *PIK3CA* mutation.** *TP53* is mutated in 16.7% of the 597 canine tumors and is the most frequently mutated gene (Fig. 5; Supplementary Data 3). Importantly, we observed a strong association between *TP53*

mutation and TMB across tumor types (Fig. 6b and Supplementary Fig. 5d). This is clearly seen in canine hemangiosarcoma and osteosarcoma, both TMB-H (Fig. 6a), and with *TP53* mutated in 59 and 50% of their tumors, respectively (Fig. 3a; Supplementary Data 3). The median TMB of osteosarcomas and hemangiosarcomas with mutant *TP53* is increased to 1.31 and 1.33, respectively, from 0.7 and 0.67 for the corresponding tumors with wild-type *TP53* (Fig. 6b). A clear association between TMB and *TP53* mutation is also noted across breeds (Fig. 7b and Supplementary Fig. 5d). Indeed, the median TMB increases with *TP53* mutation in Golden Retriever (0.53–1.4), Maltese (0.34–0.93), Greyhound (0.7–1.25), and Rottweiler (0.78–0.96) (Fig. 7b).

*PIK3CA* is the second most frequently mutated gene, mutated in 16.4% of the tumors (Fig. 5). However, in contrast to *TP53*, we did not observe a strong association (*p* < 0.05 and median fold change > 1.5) between TMB and *PIK3CA* mutation in any tumor type or breed (Figs. 6b, 7b, and Supplementary Fig. 5d).

To unbiasedly screen the association between individual gene mutation and TMB, we studied all 104 genes that are mutated in ≥5 tumors in a tumor type or breed (which can be detected with a power >0.9; see Supplementary Fig. 5a). We determined the association within each tumor type as shown in Fig. 6a, as well as within a breed after normalizing each TMB value with its tumor type median. In both analyses, *TP53* remains the most significant gene across most tumor types and breeds (Supplementary Data 5). The study also identified other genes with significant association within at least one tumor type or breed, including *ASPM*, which functions in the mitotic spindle, and *SPEF2* and *FSIP2*, both related to spermatogenesis. Notably, many of these genes are mutually inclusive with *TP53* in mutation (Supplementary Data 5).

At the pathway level, TMB is consistently associated with p53 pathway alteration (Supplementary Data 5). The cell cycle is another pathway with the association found (Supplementary Data 5).

Importantly, we also observed the association of TMB with *TP53* mutation, but not with *PIK3CA* mutation, in corresponding human adult or pediatric cancers (Fig. 6c). Moreover, in breast cancer and pediatric cancer, *TP53* has the most significant association revealed by unbiased screen (Supplementary Data 5). These findings support the dog–human homology.

**Osteosarcoma in Golden Retriever has a TMB-associated signature.** We identified three mutation signatures in the CDS regions with the WES dataset and named them S1–S3 (Fig. 8a; Supplementary Data 6). S2, matching the aging signature

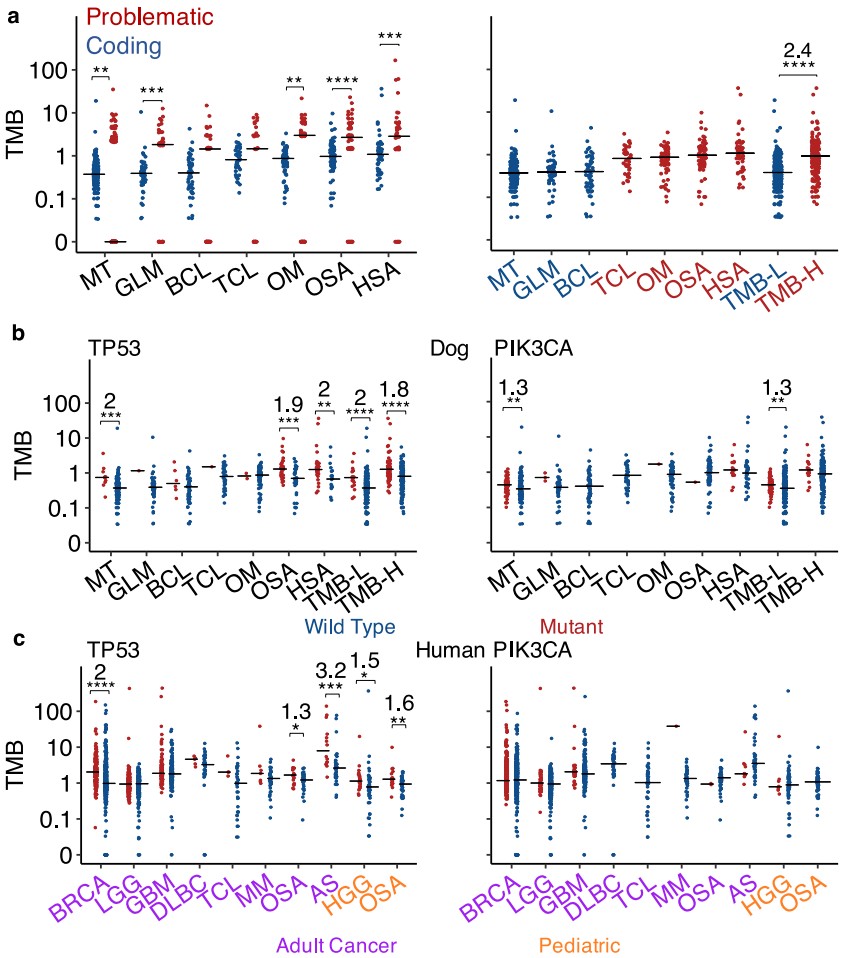

**Fig. 6 TMB varies among tumor types and is correlated with *TP53* mutation. a** TMB distributions of each canine tumor type ordered left to right from lowest to highest median values. The left plot shows that problematic genes (see "Methods") have significantly higher TMB than other genes, and thus were excluded from further analyses. The right plot indicates that canine tumors are classified into TMB-low (TMB-L) and -high (TMB-H). Two-sided Wilcoxon tests were conducted to examine the TMB difference between two groups indicated, with **, ***, **, ****, ***, and **** from left to right representing unadjusted $p = 0.006$, 0.0006, 0.001, 4.7e−6, 0.0003 and 2e−16, respectively. For significant comparisons, the fold change in median TMB is also indicated. $n = 202$, 49, 55, 38, 71, 78, 49, 306 and 236 independent tumors for each tumor type listed from left to right, respectively. **b**, **c** TMB distributions of cases with wild type (blue) or mutant (red) *TP53* or *PIK3CA* within each canine (**b**) or human (**c**) tumor type. For tumor types with both wild type and mutant groups having ≥5 tumors, two-sided Wilcoxon tests were conducted to determine the significance of the association between TMB and *TP53* or *PI3KCA*, with unadjusted p-values and fold changes shown as described in (**a**). LGG low-grade glioma, GBM glioblastoma, HGG high-grade glioma. From left to right: **b** $n = 11$, 191, 1, 49, 6, 49, 1, 37, 2, 69, 40, 38, 29, 20, 18, 288, 72 and 164 independent tumors for *TP53* (***, ***, **, ****, and **** representing $p = 0.0002$, 0.0003, 0.005, 2.4e−7 and 9.9e−5, respectively), while $n = 74$, 128, 3, 46, 55, 38, 1, 70, 1, 77, 15, 34, 77, 229, 17 and 219 independent tumors for *PIK3CA* (** and ** representing $p = 0.005$ and 0.007, respectively). **c** $n = 345$, 703, 249, 265, 121, 268, 5, 36, 4, 38, 7, 39, 20, 29, 14, 34, 22, 44, 19 and 37 independent tumors for *TP53* (****, *, ***, *, and ** representing $p = 2e−16$, 0.02, 0.0006, 0.02 and 0.008, respectively), while $n = 347$, 701, 42, 472, 36, 353, 41, 42, 1, 45, 2, 47, 10, 38, 8, 58 and 56 independent tumors for *PIK3CA*. Source data are provided as a Source Data file.

reported in human adult and pediatric cancers[6–8], is the dominant signature across tumor types and breeds (Fig. 8a). S3 matches the human UV signature and is mostly enriched in tumors of unknown tumor type (Fig. 8a). S1 lacks significant matches to any known signatures reported in human cancer (Fig. 8a). Notably, S1 is significantly enriched only in osteosarcomas of Golden Retriever dogs (Fig. 8a), and is therefore breed- and cancer-specific.

S1 is associated with TMB, but not with *TP53* mutation (Fig. 8b and Supplementary Fig. 8a). Via a systematic search, we identified four somatic mutations that are significantly associated with S1 (Supplementary Data 6). One of them is V771G of BRPF1, a subunit of the MOZ/MORF histone acetyltransferase complexes, which remodel chromatin, regulate gene expression and are implicated in cancer development[35].

We also examined the mutation signatures in noncoding regions, using 36 tumors from the WGS dataset (Supplementary Data 6). A total of five mutation signatures were identified, three of which significantly match the aging signature, COSMIC mutation signature five and the defective mismatch repair signature (MMR) reported in human tumors[6–8] (Supplementary Fig. 8b). The defective MMR signature is mostly enriched in oral melanoma (Supplementary Fig. 8b). S1 signature (Fig. 8) was not detected in these oral/ocular melanomas or glioma tumors.

## Discussion

Taking advantage of public canine data, we have investigated 684 canine cases of over 7 tumor types and over 35 breeds that are common in dogs. To our knowledge, this represents an initial pan-tumor and pan-breed study for the dog. We have built pipelines for

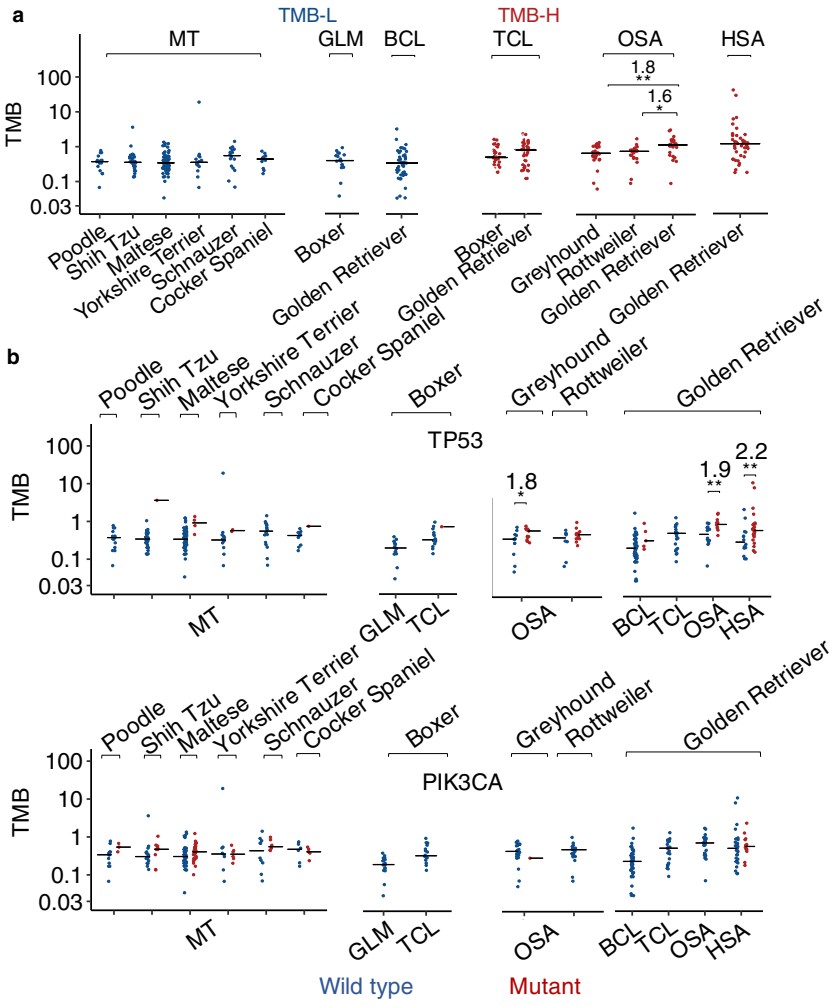

**Fig. 7 TMB is largely independent of breeds. a** TMB distributions of cases grouped by tumor type and then breed. Only groups with ≥10 tumors are shown, with n = 15, 28, 70, 14, 16, 11, 17, 44, 16, 21, 25, 21, 25 and 42 independent tumors from left to right. Two-sided Wilcoxon tests were conducted, with ** and * representing unadjusted p = 0.009 and 0.01, respectively, and fold-changes shown. **b** TMB distributions of tumors grouped by breed, tumor type, and finally *TP53* (top) or *PIK3CA* (bottom) mutation status. Only groups with *TP53* (or *PIK3CA*) wild-type and mutant-combined tumors of ≥10 are shown, with n = 15, 27, 1, 66, 4, 12, 2, 16, 10, 1, 17, 15, 1, 12, 13, 8, 13, 39, 5, 21, 14, 11, 14 and 28 (top) and n = 12, 3, 18, 10, 41, 29, 8, 6, 10, 6, 6, 5, 17, 16, 24, 1, 21, 44, 21, 25, 29 and 13 (bottom) independent tumors left to the right. Two-sided Wilcoxon tests were conducted, with *, **, and ** from left to right representing unadjusted p = 0.04, 0.003, and 0.008, respectively, and fold-changes shown. Source data are provided as a Source Data file.

comprehensive sequence QC, breed validation, and artifact reduction in somatic mutation discovery. Importantly, our work answers several important questions regarding canine tumor mutation, leading to the more precise use of the canine model in cancer research (e.g., tumor type, but not breed, should be a primary factor to consider in mutation-targeting therapy trials).

**Canine tumor alteration landscape, TMB, and *TP53*.** Our study indicates that canine somatic alteration landscape is tumor type-dependent, but largely breed-independent, for the tumor types and breeds examined here. Each of the seven canine tumor types harbors distinct gene mutations and copy number alterations. The difference is especially evident among adenoma/carcinoma, sarcoma and lymphoma. Moreover, the alteration landscape is similar among different breeds within the same tumor type but differs among different tumor types from the same breed.

Canine TMB differs among adenoma/carcinoma, sarcoma and other tumor types; however, it generally does not vary with the breed for those examined. Loss of function of TP53 is a potential reason. Canine osteosarcoma, hemangiosarcoma and oral melanoma harbor higher TMB, as well as frequent *TP53* mutation or

*MDM2* amplification (which promotes TP53 protein degradation). In contrast, canine mammary tumors and glioma harbor infrequent *TP53* mutation and lower TMB. Moreover, *TP53* mutation is strongly associated with TMB across tumor types and breeds, a pattern not observed for *PIK3CA*, the second most frequently mutated gene after *TP53*.

We propose that these observations are related to the cells of origin and tumorigenesis mechanisms, as discussed below.

Mammary adenomas or carcinomas originate from epithelial cells. The establishment of epithelial cell apical-basolateral polarity decreases cell proliferation and invasiveness, acting as a potent tumor suppressor[36–38]. PIK3CA H1047R mutation, common in canine mammary tumors, increases cell stemness[39] and decreases epithelial cell polarity, leading to accelerated cell proliferation and tumorigenesis. However, even with accelerated cell proliferation, the cell cycle checkpoint is functional and DNA damage can be repaired. This leads to a slower accumulation of mutations in the genome and lower TMB.

Osteosarcoma and oral melanoma arise from mesenchymal cells, which lack cell polarity and cell adhesion. Loss of function of TP53, due to either *TP53* mutation or *MDM2* amplification,

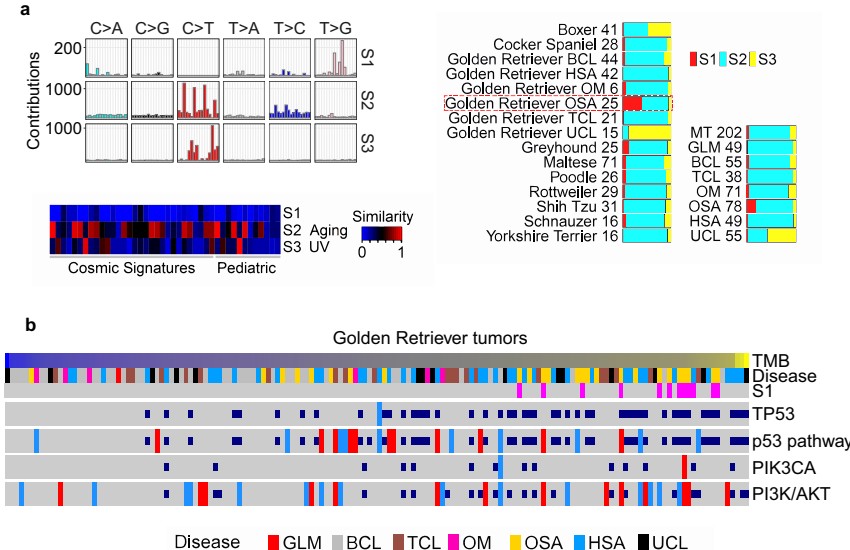

**Fig. 8 Golden Retrievers harbor a unique mutation signature that is osteosarcoma-enriched, TMB-associated, and TP53 mutation-independent.**
**a** Three mutation signatures were detected in CDS regions of 597 canine tumors from the WES dataset. Right bars indicate the distribution of the three signatures in each tumor type and each validated breed, with the numbers denoting the tumor counts. Left plots indicate the 96 base substitution patterns[6] (top) and the cosine similarity scores between each canine signature and each of the 30 COSMIC[6] and 12 pediatric[7,8] signatures (bottom). **b** Golden Retriever-specific oncoprint, including 154 animals and presented as described in Fig. 5. Source data are provided as a Source Data file.

leads to defective cell cycle checkpoints and accelerated cell cycle. As a result, fewer DNA damages are repaired and fewer DNA replication errors are corrected[40], leading to rapid accumulation of mutations in the genome and higher TMB.

In supporting the hypothesis above, we have noted a strong association between cell cycle gene alteration and TMB. However, further experimental and computational analyses are required to test this hypothesis.

Compared to B-cell lymphomas, T-cell lymphomas harbor more somatic base substitutions that have low MAF and are more random, resulting in a higher TMB using our main mutation discovery pipeline. More studies are required to understand this.

**Golden Retriever.** Exceptions to some of the general conclusions described above are noted in Golden Retriever, the largest breed in our study, with 154 animals (after QC and breed validation and prediction) and constituting all tumor types examined except for mammary tumor. For example, in osteosarcoma, TMB of Golden Retriever is significantly higher than that of other breeds, which is likely linked to a unique mutation signature enriched only in Golden Retriever. Other prominent features of Golden Retriever include higher TMB and frequent *TP53* mutation. These may be among the reasons why Golden Retriever dogs are predisposed to the development of non-epithelial cancers such as lymphoma, osteosarcoma and hemangiosarcoma. This of course needs further studies.

**Dog–human comparison.** Our pan-cancer study reveals dog–human homology in the alteration landscape. Each canine tumor type shares many of the major pathways and gene alterations with its human counterpart. However, certain differences are also noted. Different subtype composition could be one reason, e.g., more frequent *ERBB2* amplification in human breast cancer may be due to more prevalent Her2-enriched subtype in humans than in dogs. Moreover, genes can be altered via other mechanisms not examined here, including epigenetic and expression alterations. Hence, future canine subtyping and dog–human subtype comparison, along with more comprehensive alteration investigation, may further increase the dog–human homology.

The dog–human homology is also seen in TMB. First, the order of canine tumor types sorted by TMB (i.e., mammary tumor < glioma < lymphoma) is the same as that of the corresponding human cancer types. Second, across tumor types in both species, TMB is strongly associated with *TP53* mutation and p53 pathway alteration, but not with *PIK3CA* mutation. This may be related to tumor cells of origin in both species, as discussed previously.

Canine TMB is overall lower than the corresponding human adult TMB but comparable to pediatric TMB. Chronological age (in clock time) may be a factor, considering the dominance of the aging mutation signature (due to deamination of cytosine) in both species. The difference in subtype composition and driver mutations is another reason, which is clearly seen in glioma where *IHD1* mutation is frequent in humans but rare in dogs. Tumor progression stage could also be a factor, as most human adult tumors[9] used in the comparison are late-stage tumors (nearly all of human breast tumors are invasive and 33% harbor *TP53* mutation, while only half of the canine mammary tumors are invasive and <5% harbor *TP53* mutation). Further studies are needed to address this TMB difference and underlying reasons.

## Methods
### Data collection
*Canine data.* Canine WES and WGS data were downloaded from the Sequence Read Archive (SRA) database, including PRJNA489159 (mammary tumor), PRJEB12081 (oral melanoma), PRJNA579792 (glioma), PRJNA552034 (hemangiosarcoma), PRJNA247493 (osteosarcoma, lymphoma, and unclassified) and others listed in Supplementary Data 1. We also obtained other information from relevant publications[11–20,24,41,42].

Canine genome canFam3.1 and gene annotation canFam3 1.99 GTF were downloaded from the Ensembl database. Known canine germline base substitutions and small indels (55,447,895 total) were combined from (1) Ensembl canine dbSNP, canFam3; (2) the DoGSD database[43] and (3) a published study[44].

*Human data.* Mutated or amplified/deleted genes in human cancers were extracted from published studies, including 996 breast cancers[26,45], 86 high-grade pediatric gliomas for the mutation landscape study (Fig. 4)[28] and 66 high-grade pediatric gliomas for the TMB analysis (Fig. 6)[29], 511 low-grade adult gliomas[45], 37 diffuse large B-cell lymphomas[27,45], 42 T-cell lymphomas[30], 59 mucosal melanomas for the mutation landscape study (Fig. 4)[31] and 46 mucosal melanomas for the TMB analysis (Fig. 6)[15], 57 pediatric osteosarcomas[32], 46 adults osteosarcomas[32] and 48 angiosarcomas[33] (Supplementary Data 4). Human TMB values were derived from

published adult[9,33,45] and pediatric[29,32] cancer studies (Supplementary Data 5). Curated canonical cancer pathway gene lists were obtained from a TCGA pan-cancer study[46].

**Canine read mapping.** Canine sequence read pairs were mapped to the canine reference genome canFam3 using BWA-aln (version 07.17)[47]. Concordantly and uniquely mapped pairs were identified based on the flag values and TAG values (with XT: AU or XT: AM) and were used to calculate the mapping rate of each sample. Such pairs with at least one read with ≥1 bp overlapping a coding sequence (CDS) region of the canFam3 1.99 GTF annotation were used to calculate the CDS-targeting rate. Mapped read coverage was obtained using GATK (version 3.8.1)[48] DepthOfCoverage, with minimum mapping quality 10 and base quality 10. Sequencing randomness was assessed with the root mean square error (RMSE) between the actual read coverage distribution in target regions and the theoretical Poisson distribution, with $\lambda$ set to the mean coverage of each sample.

**Read harmonization.** Harmonized reads were generated from cases with matched tumor and normal samples both passing our comprehensive QC measures indicated in Fig. 1 and Supplementary Fig. 1. Briefly, for each sample, read-pairs that are concordantly mapped to the canFam3 genome were selected and processed with SAMTools (version 1.9) and then subjected to de-duplication using Picard (version 2.16.0) and indel-realignment using GATK (version 3.8.1)[48].

**Germline base substitution and small indel calling.** Germline base substitutions and small indels were first called by applying GATK 3.8.1 HaplotypeCaller to the harmonized read bam files of individual tumor or normal samples with parameters of dontUseSoftClippedBases -stand_call_conf 20.0. Variants were then filtered with GATK VariantFiltration with parameters of FS > 30.0 and QD < 2.0. Furthermore, variants with total read coverage <10 were excluded. Only germline base substitutions and small indels that were detected in both tumor and normal samples of at least one case were used for further analyses below.

**Tumor-normal sample pairing accuracy.** For a given study, let $T$ and $N$ be the total number of germline base substitutions and small indels in a tumor and normal sample, respectively, and $S$ be the total number of those shared between $T$ and $N$. The shared fraction between the tumor sample of case $i$ ($T_i$) and the normal sample of case $j$ ($N_j$) is given by $F_{i,j} = \frac{S_{i,j}}{\min(T_i, N_j)}$. When $i = j$, "self" fraction $\left(\text{Self}_i\right)$ is obtained. When $i \neq j$, "nonself" fraction is obtained. For a given case $i$, its best nonself match is identified by $\text{Best nonself}_i = \max\left(F_{i,j}, F_{j,i}\right), \forall j \in [1, n]$ and $j \neq i$, where $n$ is the total case number of the study. Thus, $\text{Self}_i - \text{Best nonself}_i$ is negative if and only if either the tumor or the normal sample of case $i$ has a better match to a sample of a different case, which indicates tumor-normal sample pairing error for case $i$.

**Breed validation and prediction.** VAF of each of the 157,628 germline base substitution and small indel variants identified as previously described was calculated in each normal sample by $\text{VAF} = \frac{\text{variant allele reads}}{\text{total reads}}$. Each variant was classified as reference (VAF < 0.2), non-reference (VAF ≥ 0.2), or not determined (ND) if total read coverage <10. Variants with ND in >20% samples were excluded, due to their poor coverage.

Only breeds with ≥10 dogs were used for breed-specific base substitution and small indel variant discovery. A variant is considered "breed-specific" if it is either breed-unique or breed-enriched. To be considered breed-unique, a variant must be (1) non-reference in ≥5 dogs of the breed; (2) non-reference in ≥40% dogs of the breed; and (3) reference in all dogs with ≥10 read coverage of the remaining breeds. Breed-enriched variants were identified with Fisher exact tests between any two breeds using the reference and non-reference sample counts. To be considered breed-enriched, a variant must be (1) enriched in breed A and (2) not enriched in any breed that is not A, against every other breed at $p \leq 0.1$.

Identified breed-specific variants were used for breed validation or prediction. First, to reduce noise, a sample to be validated or predicted should have >80% of the combined breed-specific variant sites with ≥10 read coverage for VAF calculation. For those sites with <10 read coverage, a random VAF value was assigned to each site. Breed validation was then achieved via standard hierarchical clustering with VAF values of breed-specific variants in the normal sample of each dog, as illustrated in Fig. 2.

**Somatic mutation calling.** Somatic mutations include somatic base substitutions and small indels, identified using the harmonized read bam files. MuTect (version 1.1.7)[49] was used to detect somatic base substitutions, with a minimum base quality of 30 and filtering known canine germline base substitutions from sources described earlier. Additional filtering steps were used to reduce artifacts. First, the results were subjected to a 5-step filtering process as described[15], which considers both total read coverage and mutant allele frequency (MAF). This effectively reduces artifacts with very low MAF including (1) C > T artifacts originated from the fixation process in FFPE samples[23]; and (2) G > T artifacts arisen from 8-oxoG DNA oxidative damage[22] in frozen samples from specific studies (Supplementary

Fig. 3). Second, the results were further filtered based on paired-read strand orientation bias[22]. Specifically, F1R2, where Illumina read 1 and read 2, respectively, align to the forward strand (F1) and the reverse strand (R2) of the reference genome and F2R1, the opposite of F1R2, were determined for each mutation. Then, Fisher exact tests were applied with F1R2 and F2R1 reference and mutant read to identify and exclude mutations with significant orientation bias ($p \leq 0.05$). Furthermore, cutoffs of ≥4 in total and ≥5% being mutant reads for both F1R2 and F2R1 reads were applied for: (1) G > T and C > A mutations in Broad frozen tumors; and (2) for C > T and G > A mutations in FFPE tumors of all studies, to further reduce paired-read strand orientation bias.

Somatic indels in CDS regions were discovered with Strelka[50]. As expected, small indels account for only 5% of the mutations (Supplementary Fig. 3d). Mutation annotation was performed with Annovar (version 2017Jul16)[51], using the canine annotation file described earlier.

**Tumor mutational burden.** TMB values were calculated by $\text{TMB} = \frac{\text{total somatic base substitutions and small indels in CDS}}{\text{total callable bases in millions in CDS}}$ for each case. Callable bases were identified with MuTect with the minimum base quality score set to 30.

**Validation with other somatic mutation calling tools.** To validate somatic mutation findings, other tools GATK4 MuTect2 (version 4.1.6)[48], Varscan2 (version 2.4.2)[52] and LoFreq (version 2.1.2)[53] were used as described[13]. Briefly, MuTect2 was run in the panel-of-normals (PON) mode, using harmonized reads of paired normal samples ($n = 591$) to create the PON file. SomaticSeq (version 3.4.1)[54] was used to find consensus mutation callings among MuTect2, Varscan2, and LoFreq.

**Canine retrogene and other problematic gene identification.** Problematic genes in the canFam3 1.99 GTF annotation file were identified after excluding mitochondrial genes. Problematic genes are defined as genes that: (1) have only an Ensembl ID and lack a gene symbol, name or other biologically meaningful description; and (2) consist of a single exon. A problematic gene is classified as a retrogene if its single exon arises from the fusion of partial or complete exons of a protein-coding gene.

**Somatic copy number alteration (CNA) identification.** VarScan (version 2.4.2)[55] was first applied on WES data of matched tumor and normal sample pairs. Then, CBS[56] (DNAcopy R package version1.6.4) was used to segment CDS regions, with the significance level set to 0.01 for change point identification, and 10,000 permutations performed for $p$ value calculation. Segments with $\left|\log_2\left(\frac{T}{N}\right)\right| > 1$ (T: tumor; N: normal) were considered CNAs and their overlapping genes were identified. Further selection was made by finding genes with CNAs also detected by another software, SEG (version 1.0.0)[24,57–59]. SEG was run after linear data transformation of the input $\log_2\left(\frac{T}{N}\right)$ data to set the genome-wide mean to 0, and with the initial segment probe number ($w$) set to 6 and the window size ($k$) set to 1000. CNAs were discovered with the minimal segment $\log_2\left(\frac{T}{N}\right)$ mean ($m$) set to 0.4 and the significant level ($q$) set to 0.01.

**Significant alteration identification and cross-species comparison.** Both MAF and sample recurrence were used to identify significant mutations and mutant genes (Fig. 3; Supplementary Data 3). Fisher exact tests were performed to first identify individual mutations that have significantly higher MAF, compared to the remaining mutations within a tumor. Among the identified mutations, two analyses were then performed. First, Fisher exact tests were used to find individual mutations that are significantly recurrent among the samples within a tumor type or breed. Such mutations could potentially be gain-of-function and genes harboring them may be oncogenes, e.g., PIK3CA H1047R. Second, to discover potential tumor suppressors, which harbor loss-of-function mutations that could occur at different loci among tumors (e.g., TP53), genes that harbor significant mutations in any tumor were identified. Then, using these genes as the background, Fisher exact tests were performed to identify mutant genes that are significantly recurrent across the samples within a tumor type or breed (Supplementary Data 3).

Amplified/deleted genes within a tumor were identified via Z-tests at $q \leq 0.01$ (see the previous section). Fisher exact tests were then used to identify those amplified/deleted genes that are significantly recurrent among samples within a tumor type or breed (Supplementary Data 3).

For dog–human comparison on gene or pathway alterations (Fig. 4), Fisher exact test was performed to compare the alteration recurrence of each gene or pathway among samples of matched tumor type between dog and human (Fig. 4).

Multiple testing correction with the Benjamini and Hochberg strategy[60] was applied on each Fisher exact test described above.

**Enrichment of gene and pathway alterations in a tumor type or breed.** Enrichment scores were determined by $-\log_{10}(q)$ and with positive values indicating enrichment and negative values indicating depletion. Each $q$ value was obtained from a Fisher exact test that compares the ratio of altered versus wild-type

tumors of a tumor type or a breed to that of the remaining tumor types or breeds and after applying multiple test corrections.

**Association between gene mutation and TMB.** For canine tumor type-specific association, genes that are mutated in ≥5 tumors in a specific tumor type (each of the 7 canine tumor types, TMB-L and TMB-H) were selected for Wilcoxon tests, using TMB without normalization. For breed-specific association, genes that are mutated in ≥5 tumors within a specific breed were selected, and Wilcoxon tests were conducted with normalized TMB values, given by normalized TMB = $\frac{TMB}{Tumor\ type\ TMB\ median}$. S1-high tumors (defined as tumors with ≥15 S1 mutations) and unclassified tumors were excluded from all analyses. Separate association analysis was conducted in Golden Retriever S1-high tumors only. Human association studies were performed for genes that are mutated in ≥20 tumors overall and ≥ 5 tumors in a specific cancer type. TMB normalization was performed for cross cancer type association determination.

**Sample size and power calculation.** The single-sample simulation (Supplementary Fig. 5a) estimates the power of detecting a mutation within a tumor type or breed based on the mutation prevalence and the sample size. Simulated curves (Supplementary Fig. 5a) were generated using the binomial density function with success probabilities $p$ ranging over 0.05, 0.10, 0.15, and 0.20, and sample sizes from $n = 1$ to $n = 60$. Specifically, the $y$-axis (power to detect a mutation) is $Pr(X > 0) = 1 - Pr(X = 0)$ for X binomial$(n, p)$.

To estimate the power of each two-sample Fisher exact test to determine mutation enrichment among tumor types or breeds (Fig. 3b), an R package "statmod"[61] (version 1.4.36) was used to perform 500 simulations at significance level $\alpha = 0.05$, actual sample size $n1$ and $n2$, as well as mutation prevalence $p1$ and $p2$. Assuming that the mutation is enriched in the first population ($p1 > p2$) and that $p2$ is the observed value of the second population, $p1$ was calculated as $p1 = p2 \times$ odds ratio. The odds ratio was set to the minimum observed value among (1) all significant comparisons (which is 2.0) or (2) gene/pathway-specific significant comparisons (Fig. 3b and Supplementary Fig. 3b), among tumor types. Moreover, the observed odds ratio was used if it is higher than the minimum odds ratio chosen above. If the calculated $p1 > 1$, then $p1 = 1$ and $p2 = \frac{p1}{odds\ ratio}$.

To estimate the power of each two-sample Wilcoxon test to determine TMB differences between tumor types or breeds (Figs. 6 and 7), Wilcoxon $p$ values were calculated on two simulated normal distributions for 10,000 simulation iterations. The simulation was performed with the actual sample size $n1$ and $n2$, $\sigma = 1$ for both groups, $\mu_{low} = 0$, and $\mu_{high}$ = standardized effect size. The standardized effect size was estimated as $0.93 = \frac{x_{high} - x_{low}}{s}$, where $x_{high}$ and $x_{low}$ are the sample average of $\log_{10}$-transformed TMB values of the TMB-H and TMB-L tumor groups (Fig. 6a), respectively, while $s$ is the pooled standard deviation.

The same strategy was used to estimate the power of Wilcoxon tests for TMB association with *TP53* and *PIK3CA* mutation analyses (Figs. 6b and 7b). The standardized effect size for canine comparisons was estimated from the *TP53* association within TMB-L (which is 0.93) and TMB-H (which is 0.76) tumors. For human comparisons, the minimum and maximum observed effect size were estimated from the TP53 association within breast tumors (0.65) and angiosarcoma tumors (1.0).

**Mutation signature identification.** Mutation signatures in coding regions were discovered with WES tumors (597 total), while mutation signatures in noncoding regions were identified with WGS tumors (36 total). The SignatureAnalyzer R software[62] was applied on the raw counts of filtered somatic mutations of these tumors, using the default parameters except for the number of iterations = 40 and hyper = False. The most frequent solution for mutation signature deconvolution has three signatures (22 of 40 iterations) in WES analyses and five signatures (37 of 40 iterations) in WGS analyses. These signatures were used for subsequent analyses. For dog–human comparison, human COSMIC signatures v2[6] and pediatric cancer signatures[7,8] were adjusted using the human genomic background trinucleotide probabilities, while canine signatures were adjusted using the canine exonic trinucleotide probabilities for coding signatures and the canine genomic background trinucleotide probabilities for non-coding signatures (Supplementary Data 6). The adjusted signatures were compared via cosine similarity.

**S1 signature-associated mutation discovery.** Among 10,040 somatic mutations identified in 597 tumors, 39 were recurrent in ≥5 tumors and were used for the association study, by comparing S1-high tumors versus S1-low tumors via Fisher exact test at false discovery rate (FDR) ≤ 0.1. This results in 11 mutations, which were further selected for evolutionary conservation.

**Reporting summary.** Further information on research design is available in the Nature Research Reporting Summary linked to this article.

## Data availability

Our study focuses on publicly available canine and human data. Canine WES and WGS data were downloaded from the Sequence Read Archive (SRA) database, including BioProject IDs PRJNA489159 (WES; mammary tumor), PRJNA552905 (WES; mammary tumor), PRJEB12081 (WES; oral melanoma), PRJNA247493 (WES; osteosarcoma, B-cell lymphoma, T-cell lymphoma and unclassified), PRJNA525883 (WES and WGS; osteosarcoma), PRJNA579792 (WES and WGS; glioma), PRJNA417727 (WES; hemangiosarcoma), PRJNA552034 (WES; hemangiosarcoma) and PRJNA389294 (WGS; melanoma), as listed in Supplementary Data 1. We also obtained case information (e.g., B-cell or T-cell lymphoma and breed data of the lymphoma study[14], tumor and normal status of the oral melanoma samples[15]) from supplementary tables and supplementary data of relevant publications[11–20]. Harmonized variant-level data are provided in Supplementary Data 3. Harmonized read-level data, which are processed data (see Methods), are provided as bam files and are 15 TB in total. Due to the large size and nature of the files, the data will be available upon request (e.g., via FTP) by contacting the corresponding author at szhao@uga.edu. Mutated or amplified/deleted genes, altered pathways and TMB in human cancers were extracted from the cBioPortal database[45] and supplementary tables and supplementary data of published studies[9,26–29,31–33], as described in the Article and Supplementary Data 4 and 5. Source data are provided with this paper.

## Code availability

The pipeline, example input files, and the manual for canine breed validation and prediction (Supplementary Software 1) have been deposited to GitHub, free to the public at https://github.com/ZhaoS-Lab/breed_prediction (https://doi.org/10.5281/zenodo.4948044)[63]. Mutation discovery and other analyses utilize published software tools, as described in the Methods section of the Article.

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

## Acknowledgements

We thank various research laboratories for publishing the canine and human data used in this study, and the Georgia Advanced Computing Resource Center (GACRC) for supporting this work. This work is funded by NCI R01 CA182093 and R01 CA252713.

## Author contributions

S.Z. conceived the study. B.A.A. developed and implemented the breed validation algorithms. B.A.A., K.L.H., J.W., Y.F., and T.W. performed the data analyses. B.A.A. and K.K.D. performed the sample size and power calculation. S.Z., B.A.A., K.L.H., J.W., Y.F. and T.W. wrote and edited the manuscript.

## Competing interests

The authors declare no competing interests.
