## [Peer Review File · Nature Communications]

REVIEWER COMMENTS

Reviewer #1 (Remarks to the Author): Expert in pan-cancer genomics

Alsaihati and colleagues have undertaken a pan-cancer, pan-breed analysis of canine tumors. They identified a series of large datasets, systematically reprocessed them, and provide comparison across breeds, cancer types, and species (with humans). The analysis is generally (very) well-performed and will be of quite significant impact, whereas the reporting has several flaws that limit the current manuscript.

1. Discovery & Validation

I greatly appreciate the author's attempt to create a validation cohort. It's a thoughtful idea, but the validation dataset size is both small and contaminated ($n=90$ WES + $n = 102$ WGS, where over half already exist in the discovery cohort). The authors should simply merge the two datasets into a single cohort, retaining WES-based coding SNV detection and WGS-based non-coding SNVs (for mutational signatures) and CNAs. This will significantly strengthen and simplify the presentation.

2. Dataset

Apologies if I missed this, but it is unclear where the output data, which is critical to both the value of this study and its interpretability, have been deposited. Clearly both the harmonized read-level and the harmonized variant-level data need to be provided in a publicly available way. The study would have been dramatically enhanced by provision of a cBioPortal interface to these data.

3. Terminology

Please be precise on terminology -- "breed-specific variants", are these SNPs? Other? Similarly "somatic mutations" appears to be used when referring to SNVs: CNAs (and GRs, and indels) are all forms of mutation. Indeed the very phrase TMB is incorrect for this reason, and should be referred to as simply "somatic SNVs/Mbp" in the figures and bulk of the text.

4. Software

Please ensure the software for SNP-based breed-assignment is made available, this is a key advance of the study, and a suitable github/equivalent repository with the software and required input files for its execution are needed.

5. Data visualization

In general I found the data-visualization sufficient, although there were many plots that used red & green together (which is not recommended in Nature style given the frequency of red-green color-

blindness). Additional, to clarify the presentation direct and simple presentation of the frequency of human vs. canine drivers should be given in a scatterplot per cancer type (points = genes, symbols = breeds).

6. Statistics

In many cases it is unavoidable that statistical power here is very limited. Nevertheless, all claims should be clearly supported with statistical phrasing -- both in terms of whether individual mutations occur above background rates, and in terms of differences between breeds and species. Given the dataset, relatively simple statistical models (i.e. controlling only for mutation load) are sufficient, but the lack of statistics hinders interpretation of much of the data. Some discussed results may not be significant after this analysis and appropriate multiple testing correction, and discussion of those should be removed or curtailed.

Reviewer #2 (Remarks to the Author): Expert in comparative cancer genomics

In their paper Alsaihati et al perform a pan-cancer analysis of published canine cancer genome studies to explore the landscape of driver genes and also mutational processes. I have several specific comments on this manuscript.

1. More information on how each animal was assigned to a breed is required in the methods. From the legend of figure 1 it appears that breeds were assigned using

germline SNPs but I'm not sure I could reproduce this analysis using the details provided. Ideally it would be useful if the code for this analysis was released into Github so it could be used by others. Not all animals will be pure bred and it is not clear how this ambiguity was handled.

2. Some of the studies included in this paper were FFPE sequencing studies and as such there are well established artefacts such as C>T and C>A that can be caused by the fixation process. It is unclear to me how these potential errors were filtered from these datasets. How did the variant calls made by the authors' pipeline compare to those made by the authors of the original papers?

3. If I understand the paper correctly, if coverage was sufficient a sample was included in the analysis of tumour mutational burden. It has previously been shown that the sensitivity of somatic variant calling is different when comparing 30x to 50x or 100x (PMID: 26647970) how might this influence the results?

4. I think the authors should state the power with which they make the statement that "Our study indicates that canine somatic mutation landscape is tumor type-dependent, but largely breed-independent.". I believe this to be a correct and logical conclusion but for some breeds the authors have very few dogs/tumours (Supp table S1). In humans we know that there are some differences (for some tumour types) in the somatic mutational landscape when tumours from different populations are compared. For example:

<https://www.biorxiv.org/content/10.1101/2020.10.28.359240v1>

The comments above reflect a need for more details in the methods so that the analysis is clear. As above it would be useful if all the scripts were released into Github. In general the studies are well performed and illustrate the value of canine cancer models. The figures are in general well presented but when they are published will be tiny. It might help if Figure 1G was a new figure on its own so that it is clearly visible.

David Adams, Sanger Institute.

Response to Reviewers

We greatly appreciate the reviewers' insightful comments, which not only guide our manuscript revision but also make us better understand cancer mutations. Summarized below are our point-to-point response, with reviewers' comments in **blue**, our response in **black** and revised text copied from the manuscript in **green**.

Reviewer #1

1. Discovery & Validation

I greatly appreciate the author's attempt to create a validation cohort. It's a thoughtful idea, but the validation dataset size is both small and contaminated (n=90 WES + n = 102 WGS, where over half already exist in the discovery cohort). The authors should simply merge the two datasets into a single cohort, retaining WES-based coding SNV detection and WGS-based non-coding SNVs (for mutational signatures) and CNAs. This will significantly strengthen and simplify the presentation.

We have followed the suggestion and first merged all WES studies, after applying the same quality control (QC) measures on each (Figure 1; Table S1A). This yields 591 canine cases (597 tumors and 591 matching normal samples) with WES data for subsequent coding mutation analysis (Table S1A). We have also addressed possible sequence coverage difference among the studies (see Response to Reviewer 2, comment 3).

We have revised paragraph 1 on page 4 to reflect this change, as pasted below.

The WES dataset consists of 1,316 paired tumor and normal samples of 654 animals from 9 Bioprojects (Table S1A). These include 204 cases (408 samples) of mammary tumor^{14,15}, 56 cases (112 samples) of glioma¹⁶, 61 cases (122 samples) of B-cell lymphoma¹⁷, 39 cases (78 samples) of T-cell lymphoma¹⁷, 65 cases (136 samples) of oral melanoma¹⁸, 78 cases (156 samples) of osteosarcoma^{19,20}, 68 cases (138 samples) of hemangiosarcoma^{21,22} and 83 cases (166 samples) of unclassified tumors (Table S1A). They represent over 35 breeds, including Golden Retriever (163 dogs), Maltese (69 dogs), Poodle (38 dogs), Boxer (36 dogs) and others listed in Table S1A.

The WGS dataset is much smaller, with only 36 cases remaining after sequence data QC and removing tumors that do not belong to any of the 7 tumor types (Table S1B). Due to the small sample size, we have used the WGS dataset only to investigate noncoding mutation signatures (Figure S8) and to validate our breed validation/prediction strategy (Figure 2).

The WGS dataset is described in paragraph 3 on page 5, as pasted below.

We also performed similar QC analyses on the WGS dataset, which consists of 172 paired tumor and normal samples from 86 animals with glioma (67 cases)¹⁶, oral or ocular melanoma (4 cases)²³, or osteosarcoma (15 cases)²⁰ (Table S1B). Close to 30 breeds are covered, including Boxer (24 animals), Boston Terrier (11 animals) and others listed in Table S1B. We found 25 samples with a mapping rate <60% and 25 samples with a sequence coverage <30X (Figure S1; Table 1B), and excluded them from further analysis. Because of the small sample size (only 72

paired tumor and normal samples from 36 cases passed QC) (Table S1B), we used the WGS dataset only for breed validation and non-coding mutation signature finding.

We have revised other relevant places throughout the manuscript to reflect these changes, which are tracked.

2. Dataset

Apologies if I missed this, but it is unclear where the output data, which is critical to both the value of this study and its interpretability, have been deposited. Clearly both the harmonized read-level and the harmonized variant-level data need to be provided in a publicly available way. The study would have been dramatically enhanced by provision of a cBioPortal interface to these data.

Harmonized variant-level data are provided in Supplementary Tables, including Table S3, while harmonized read-level data are available upon request. This is described under “**Data Availability**” on page 17, as pasted below.

Our study focuses on publicly available canine and human data. Canine WES and WGS data were downloaded from the Sequence Read Archive (SRA) database, as listed in Table S1A. We also obtained case information from relevant publications^{14-23,27,44,45}. Harmonized variant-level data are provided in Supplementary Tables. Due to their large size, harmonized read-level data cannot be submitted as Supplementary Data; however they are available upon request.

Mutated or amplified/deleted genes, and altered pathways in human cancers were extracted from published studies^{10-12,29-32,34-36,48}, as described in the Article and Supplementary Tables.

Source data are provided with this paper, in relevant Supplementary Tables.

We have contacted the cBioPortal team, but they are currently unable to host any nonhuman data. We plan to build a cBioportal interface for the dog. It however will take time, because our team is small and Dr. Alsaihati, the first author, and Dr. Wang, another co-author, have recently graduated and left the lab.

3. Terminology

Please be precise on terminology -- "breed-specific variants", are these SNPs? Other? Similarly "somatic mutations" appears to be used when referring to SNVs: CNAs (and GRs, and indels) are all forms of mutation. Indeed the very phrase TMB is incorrect for this reason, and should be referred to as simply "somatic SNVs/Mbp" in the figures and bulk of the text.

We apologize for the confusion. To correct this, we have defined each term when it first appears in the manuscript.

Breed-specific variants include germline base substitutions and small indels, as described in paragraph 4 on page 5 and pasted below.

To assess the breed data accuracy, we focused on the 10 pure breeds in the WES dataset with each having ≥ 10 animals passing QC measures specified in Figure 1. We identified 5,363 breed-specific variants (Table S2), defined as germline base substitutions and small indels that are unique to or enriched in one of these breeds.

In our study, somatic mutations include somatic base substitutions and small indels, as defined in the last paragraph on page 3 and pasted below.

We then investigated somatic mutations, which include somatic base substitutions and small indels, as well as gene amplifications and deletions in 597 tumors from 591 cases with WES data passing our QC measures.

We understand that CNAs (studied here) represent another form of mutation. The reason why we do not include CNAs in somatic mutations is our extensive use of data from the cBioportal interface, where mutations and CNAs are listed separately. We have adopted the same strategy as cBioportal to simplify the presentation, as exemplified by TMB below.

In our study, TMB is defined as “the number of somatic base substitutions and small indels per Mb callable coding sequence (CDS)” (from the last paragraph under Introduction, page 3). We agree with the reviewer that this definition is not precise, but it is frequently used in literature at present (e.g., references 11 and 36). As small indels are also included in the calculation, we have chosen to keep the use of TMB, instead of somatic SNVs/Mbp.

To avoid confusion, we have changed “mutation landscape” to “alteration landscape”, which consists of “genes that harbor somatic non-synonymous base substitutions or small indels, as well as genes that are amplified or deleted” as described in paragraph 5 on page 6.

We have revised other relevant places throughout the manuscript for the consistent use of these terms.

4. Software

Please ensure the software for SNP-based breed-assignment is made available, this is a key advance of the study, and a suitable github/equivalent repository with the software and required input files for its execution are needed.

The pipeline, example input files, and the manual for canine breed validation and prediction have been deposited to GitHub, free to the public at https://github.com/ZhaoS-Lab/breed_prediction. This is stated under “**Code Availability**” on page 17.

5. Data visualization

In general I found the data-visualization sufficient, although there were many plots that used red & green together (which is not recommended in Nature style given the frequency of red-green color-blindness). Additional, to clarify the presentation direct and simple presentation of the frequency of human vs. canine drivers should be given in a scatterplot per cancer type (points = genes, symbols = breeds).

We have revised the figures to avoid using red & green together in any plots.

Following the recommendation, we have used a scatter plot (Figure 4) to present the alteration frequency of human vs. canine drivers for each tumor type. The findings are described in detail under the “**Canine and human tumors share many major alterations**” section on pages 7-8. Briefly, significantly mutated genes, along with curated cancer pathways, are statistically compared between two species. Their breed enrichment is also examined.

6. Statistics

In many cases it is unavoidable that statistical power here is very limited. Nevertheless, all claims should be clearly supported with statistical phrasing -- both in terms of whether individual mutations occur above background rates, and in terms of differences between breeds and species. Given the dataset, relatively simple statistical models (i.e. controlling only for mutation load) are sufficient, but the lack of statistics hinders interpretation of much of the data. Some discussed results may not be significant after this analysis and appropriate multiple testing correction, and discussion of those should be removed or curtailed.

We have performed a number of analyses to identified significant mutations, as described under the “**Significant alteration identification and cross-species comparison**” section on page 15, as pasted below.

Both MAF and sample recurrence were used to identify significant mutations and mutant genes (Figure 3; Table S3). Fisher exact tests were performed to first identify individual mutations that have significantly higher MAF compared to the remaining mutations within a tumor. Among the identified mutations, two analyses were then performed. First, Fisher exact tests were used to find individual mutations that are significantly recurrent among the samples within a tumor type or breed. Such mutations could potentially be gain-of-function and genes harboring them may be oncogenes, e.g., PIK3CA H1047R. Second, to discover potential tumor suppressors, which harbor loss-of-function mutations that could occur at different loci among tumors (e.g., *TP53*), genes that harbor significant mutations in any tumor were identified. Then, using these genes as the background, Fisher exact tests were performed to identify mutant genes that are significantly recurrent across the samples within a tumor type or breed (Table S3).

Amplified/deleted genes within a tumor were identified via Z-tests at $q \leq 0.01$, as described previously. Fisher exact tests were then used to identify those amplified/deleted genes that are significantly recurrent among samples within a tumor type or breed (Table S3).

For dog-human comparison on gene or pathway alterations (Figure 4), Fisher exact test was performed to compare the alteration recurrence of each gene or pathway among samples of matched tumor type between dog and human (Figure 4).

Multiple testing correction with the Benjamini and Hochberg strategy⁶³ was applied on each Fisher exact test described above.

For cross canine tumor type or breed comparison, we conducted Fisher exact tests as shown Figure 3b. For TMB analyses, we conducted Wilcoxon tests, as indicated in Figures 6 and 7.

We have also provided power calculation for each test (see Response to Reviewer 2, comment 3).

We have revised relevant places throughout the manuscript to reflect these statistical analyses, which are also discussed in our Response to Reviewer 2 below.

Reviewer #2

1. More information on how each animal was assigned to a breed is required in the methods. From the legend of figure 1 it appears that breeds were assigned using germline SNPs but I'm not sure I could reproduce this analysis using the details provided. Ideally it would be useful if the code for this analysis was released into Github so it could be used by others. Not all animals will be pure bred and it is not clear how this ambiguity was handled.

The software, example input files, and the manual have been deposited to GitHub (see Response to Reviewer 1, comment 4).

Our method works only for pure breeds, and mixed breeds cannot be assigned. Assuming that the majority of pure breed dogs from the original studies are correctly assigned, we used these dogs to identify breed-specific germline base substitutions and small indels (see Response to Reviewer 1, comment 3). When clustering with their variant allele frequency (VAF) values, each pure breed displays a distinct pattern (Figures 2 and S2). However, mixed breed dogs lack any such patterns, as shown in Figure S2c which includes all 24 mixed breed dogs from the original studies. Furthermore, our analyses indicate that 5 out of the 395 original pure breed dogs lack specific patterns (Figure 2). These 5 dogs may possibly be mixed breeds or other breeds not examined here, and were reclassified as “unknown” (Figure 2). These results are described under “**Breed-specific germline analysis for breed validation**” on pages 5-6.

The method is described in detail under the “**Breed validation and prediction**” section on page 13, as pasted below.

Variant allele frequency (VAF) of each of the 157,628 germline base substitution and small indel variants identified as previously described was calculated in each normal sample by $VAF = \frac{\text{variant allele reads}}{\text{total reads}}$. Each variant was classified as reference (VAF <0.2), non-reference (VAF ≥0.2), or not determined (ND) if total read coverage <10. Variants with ND in >20% samples were excluded, due to their poor coverage.

Only breeds with ≥10 dogs were used for breed-specific base substitution and small indel variant discovery. A variant is considered “breed-specific” if it is either breed-unique or breed-enriched. To be considered breed-unique, a variant must be: 1) non-reference in ≥5 dogs of the breed; 2) non-reference in ≥40% dogs of the breed; and 3) reference in all dogs with ≥10 read coverage of the remaining breeds. Breed-enriched variants were identified with Fisher exact tests between any two breeds using the reference and non-reference sample counts. To be considered breed-enriched, a variant must be 1) enriched in breed A and 2) not enriched in any breed that is not A, against every other breed at $P \leq 0.1$.

Identified breed-specific variants were used for breed validation or prediction. First, to reduce noise, a sample to be validated or predicted should have >80% of the combined breed-specific variant sites with ≥ 10 read coverage for VAF calculation. For those sites with <10 read coverage, a random VAF value was assigned to each site. Breed validation was then achieved via standard hierarchical clustering with VAF values of breed-specific variants in the normal sample of each dog, as illustrated in Figure 2.

2. Some of the studies included in this paper were FFPE sequencing studies and as such there are well established artefacts such as C>T and C>A that can be caused by the fixation process. It is unclear to me how these potential errors were filtered from these datasets. How did the variant calls made by the authors' pipeline compare to those made by the authors of the original papers?

We appreciate the comment and have improved our mutation discovery pipeline, as summarized in Figure S3.

The studies indeed consist of both frozen and FFPE samples (Table S1). To address the mutation artifacts in FFPE samples, we first adopted the 5-step filtering strategy described in the original oral melanoma paper¹⁸, which considers both mutant allele frequency (MAF) and read coverage. This effectively reduces artifacts, including C>T mutations with very low MAF values in FFPE samples (Figures S3a and S3b), as well as C>A/G>T artifacts²⁵ in frozen tumors from studies by Broad Institute (Figures S3a and S3b).

However, some artifacts yet remain. These include: 1) a higher C>T/G>A mutation rate in FFPE samples than in frozen samples within the same study (see glioma and hemangiosarcoma studies in Figure S3b); and 2) more frequent C>A/G>T mutations with MAF <0.1 in frozen samples of Broad studies (Figure S3b).

To further reduce these artifacts, we investigated paired-read strand orientation bias, such as that arisen from 8-oxoG DNA oxidative damage, where G>T is always found in Illumina read 1 while C>A is always found in Illumina read 2²⁵. We developed further filtering steps by using Fisher exact test and mutant read cutoffs in both read strand orientations (see below). This further reduces C>A/G>T artifacts in Broad frozen samples, as well as artifacts in FFPE samples (Figure S3c). As a result, the total numbers of mutations between FFPE and frozen samples are more comparable, especially in the glioma study (Figure S3c).

This mutation filtering strategy is described under the “**Somatic mutation calling**” section on page 14, as pasted below.

Additional filtering steps were used to reduce artifacts. First, the results were subjected to a 5-step filtering process as described¹⁸, which considers both total read coverage and mutant allele frequency (MAF). This effectively reduces artifacts with very low MAF including: 1) C>T artifacts originated from the fixation process in FFPE samples²⁶; and 2) G>T artifacts arisen from 8-oxoG DNA oxidative damage²⁵ in frozen samples from specific studies (Figure S3; Table S3A). Second, the results were further filtered based on paired-read strand orientation bias²⁵. Specifically, F1R2, where Illumina read 1 and read 2 respectively align to the forward strand

(F1) and the reverse strand (R2) of the reference genome, and F2R1, the opposite of F1R2, were determined for each mutation. Then, Fisher exact tests were applied with F1R2 and F2R1 reference and mutant reads to identify and exclude mutations with significant orientation bias ($P \leq 0.05$). Furthermore, cutoffs of ≥ 4 in total and $\geq 5\%$ being mutant reads for both F1R2 and F2R1 reads were applied for: 1) G>T and C>A mutations in Broad frozen tumors; and 2) for C>T and G>A mutations in FFPE tumors of all studies, to further reduce paired-read strand orientation bias.

We have compared our mutations to those reported in the original publications (Figure S4), as described in paragraph 4 on page 6 and pasted below.

We compared each mutation in each tumor between our study and the original publications, including the genomic coordinate and the actual mutation, which are published only for the mammary tumor¹⁴ and oral melanoma¹⁸ studies. For oral melanoma, we found a median overlap rate of 67% with 5-step filtering and of 59% with further paired-read strand orientation filtering (Figure S4b; Table S4). We manually examined >20 mutations detected only by our pipeline or in the original publications, and found that all appear to be valid base changes (a few examples provided in Figure S4). Thus, the difference is likely due to variations in read cleaning, germline mutation filtering and artifact filtering. For mammary tumor, the overlap rate is lower (43%) (Figure S4c; Table S4) due to different mutation calling software, as 66% overlap was achieved when we used MuTect2 as in the original publication¹⁴ (Figure S4c).

3. If I understand the paper correctly, if coverage was sufficient a sample was included in the analysis of tumour mutational burden. It has previously been shown that the sensitivity of somatic variant calling is different when comparing 30x to 50x or 100x (PMID: 26647970) how might this influence the results?

To address this issue, we divided the samples into 30-50x, 50-100x and $\geq 100x$ coverage groups within each tumor type (note that our QC pipeline excludes samples with coverage $< 30x$; see Figure 1). We then compared the TMB across tumor types for groups with matched coverage (Figure S6a and S6b), which supports our original conclusion that TMB is tumor type-dependent. This result is described in paragraph 7 on page 8, as pasted below.

As sequence coverage influences the sensitivity of somatic mutation discovery³⁷, we performed TMB comparison across tumor types controlling for sequence coverage (at 30-50x, 50-100x and $> 100x$) (Figure S6a). The analysis confirms our original conclusion that TMB is tumor type-dependent (Figure S6b).

The coverage difference should not affect our conclusion regarding breed independence of TMB (Figure 7). This analysis is performed within each tumor type, where no significant difference in sequence coverage was observed among the breeds except for Schnauzer which has a small sample size (Figure S6c).

4. I think the authors should state the power with which they make the statement that “Our study indicates that canine somatic mutation landscape is tumor type-dependent, but largely breed-independent.”. I believe this to be a correct and logical conclusion but for some breeds the

authors have very few dogs/tumours (Supp table S1). In humans we know that there are some differences (for some tumour types) in the somatic mutational landscape when tumours from different populations are compared. For example: <https://www.biorxiv.org/content/10.1101/2020.10.28.359240v1>

We have performed power calculations for the alteration landscape analyses shown in Figure 3 (note that “mutation landscape” is revised to “alteration landscape”, as explained in our Response to Reviewer 1, comment 3). The powers are indicated in Figure S5.

First, we determined the power for individual alteration discovery (Figure S5a), as described in paragraph 5 on page 6 and pasted below.

We identified genes that harbor somatic non-synonymous base substitutions or small indels, as well as genes that are amplified or deleted, in each tumor (Table S3). We then examined the alteration landscape (Figure 3a), which consists of these altered genes that can be detected at ≥ 0.8 power within a tumor type or a breed based on our sample size calculation (Figure S5a).

Second, for alteration landscape comparison among tumor types or breeds (Figure 3b), we only used the most recurrently altered genes and pathways. Our two-sample power calculation (Figure S5b) indicates a power of 0.8-1.0 for these alterations for: 1) the majority of tumor type comparisons; and 2) the breed comparisons within at least one tumor type. We have modified our conclusion, as described in paragraph 4 on page 7 and pasted below.

In contrast to tumor type, canine alteration landscape appears largely breed-independent among the breeds examined (Figure 3a). To statistically test this, we performed Fisher exact tests on the most recurrently altered genes (*TP53*, *PIK3CA* and *CDKN2A*) and pathways (p53, PI3K, cell cycle and RTK/RAS) to achieve a larger power (Figure S3b). Most of these alterations do not differ significantly in their enrichment or depletion levels among different breeds within the same tumor type, unlike the tumor type comparison (Figures 3b; Table S3). For example, mammary tumors of Maltese, Shih Tzu and Yorkshire Terrier dogs all have frequent *PIK3CA* mutation and PI3K pathway alteration (Figures 3b and S3b). However, various tumor types of Golden Retriever dogs differ significantly in these alterations (Figures 3b and S3b).

The power analysis is described under the “**Sample size and power calculation**” section on page 16, as pasted below.

The single-sample simulation (Figure S5a) estimates the power of detecting a mutation within a tumor type or breed based on the mutation prevalence and the sample size. Simulated curves (Figure S5a) were generated using the binomial density function with success probabilities p ranging over 0.05, 0.10, 0.15 and 0.20, and sample sizes from $n = 1$ to $n = 60$. Specifically, the y-axis (power to detect a mutation) is $\Pr(X > 0) = 1 - \Pr(X = 0)$ for X binomial(n,p).

To estimate the power of each two-sample Fisher exact test to determine mutation enrichment among tumor types or breeds (Figure 3b), an R package “statmod”⁶⁴ was used to perform 500 simulations at significance level $\alpha = 0.05$, actual sample size n_1 and n_2 , as well as mutation prevalence p_1 and p_2 . Assuming that the mutation is enriched in the first population ($p_1 > p_2$)

and that p_2 is the observed value of the second population, p_1 was calculated as $p_1 = p_2 \times$ odds ratio. The odds ratio was set to the minimum observed value among 1) all significant comparisons (which is 2.0) or 2) gene/pathway-specific significant comparisons (Figures 3b and S3b), among tumor types. Moreover, the observed odds ratio was used if it is higher than the minimum odds ratio chosen above. If the calculated $p_1 > 1$, then $p_1 = 1$ and $p_2 = \frac{p_1}{odds\ ratio}$.

The comments above reflect a need for more details in the methods so that the analysis is clear. As above it would be useful if all the scripts were released into Github.

See our response to comment 1, and the “**Code Availability**” statement on page 17.

In general the studies are well performed and illustrate the value of canine cancer models. The figures are in general well presented but when they are published will be tiny. It might help if Figure 1G was a new figure on its own so that it is clearly visible.

Figure 1G is now Figure 2. Moreover, to enlarge the text font, we have broken each of the original Figures 2 and 5 into two figures, which are Figures 3 and 4 (alteration landscape and dog-human comparison respectively) and Figures 7 and 8 (canine breed TMB comparison and mutation signature respectively) in the current submission.

REVIEWERS' COMMENTS

Reviewer #1 (Remarks to the Author):

Thank you for addressing all my questions

Reviewer #2 (Remarks to the Author):

I am happy with the thoughtful review of the manuscript provided by the authors. I have one general comment about data sharing. Some of the "ReadsData" has been shared using Google Drive. This is not a long term stable way of sharing data. I would suggest that these data are added to Github or FigShare where they will have a DOI assigned.

Response to Reviewers

Reviewer #2 comment: I have one general comment about data sharing. Some of the "ReadsData" has been shared using Google Drive. This is not a long term stable way of sharing data. I would suggest that these data are added to Github or FigShare where they will have a DOI assigned.

Our response: Our harmonised read data are about 15 TB in total. GitHub, Zenodo and other places which we have checked are unable to publish such large and reprocessed (not original) data. It costs about \$38,000 for FigShare to publish the data, a price which we unfortunately cannot afford. After consulting with the editor Dr. Gutierrez, we will

- a) provide the data upon request (e.g., via FTP), as described in our Data Availability statement on page 17 of the manuscript; and
- b) work with the NCI ICDC database team to permanently publish these data at <https://caninecommons.cancer.gov/#>, which however will take several months. Once the data is hosted at ICDC, we will provide the link at the comment section at our published paper site.